# Intragenic *EGFR::EGFR*.E1E8 Fusion (EGFRvIII) in 4331 Solid Tumors

**DOI:** 10.3390/cancers16010006

**Published:** 2023-12-19

**Authors:** Lan Zheng, Rajyalakshmi Luthra, Hector A. Alvarez, F. Anthony San Lucas, Dzifa Y. Duose, Ignacio I. Wistuba, Gregory N. Fuller, Leomar Y. Ballester, Sinchita Roy-Chowdhuri, Keith J. Sweeney, Asif Rashid, Richard K. Yang, Wei Chen, Audrey Liu, Yun Wu, Constance Albarracin, Keyur P. Patel, Mark J. Routbort, Aysegul A. Sahin, Qingqing Ding, Hui Chen

**Affiliations:** 1Department of Pathology, The University of Texas MD Anderson Cancer Center, Houston, TX 77030, USAyunwu@mdanderson.org (Y.W.); calbarra@mdanderson.org (C.A.); asahin@mdanderson.org (A.A.S.); 2Department of Hematopathology, The University of Texas MD Anderson Cancer Center, Houston, TX 77030, USAhaalvarez@mdanderson.org (H.A.A.);; 3Department of Translational Molecular Pathology, The University of Texas MD Anderson Cancer Center, Houston, TX 77030, USAiiwistuba@mdanderson.org (I.I.W.)

**Keywords:** EGFRvIII, *EGFR::EGFR*.E1E8 fusion, glioblastoma, breast sarcomatoid neoplasm, malignant phyllodes tumor

## Abstract

**Simple Summary:**

Epidermal growth factor receptor variant III (EGFRvIII) is caused by the deletion of six exons and the fusion of exons 1 to exon 8. EGFRvIII occurs frequently in glioblastoma, a type of high-grade brain tumor; however, its presence in other solid tumors is not well characterized. Upon reviewing 4331 solid tumor cases tested via the 610-gene sequencing platform, EGFRvIII was identified in 17 cases, including 16 brain tumors and one breast tumor. EGFRvIII-positive brain tumors were all glioblastoma with wild-type *IDH1/2* status, most with *EGFR* amplification and *EGFR* mutation. The only EGFRvIII-positive breast lesion was in a young female patient. A separate breast case tested outside our institution with reported EGFRvIII was noted in a young female patient. Both EGFRvIII-positive breast tumors showed high-grade sarcomatoid morphology. In summary, EGFRvIII is rare, occurring primarily in glioblastoma and rarely in breast sarcomatoid neoplasm. This select group of patients may benefit from chemotherapy and/or targeted anti-EGFR therapy.

**Abstract:**

Epidermal growth factor receptor variant III (EGFRvIII, the deletion of exons 2–7) is a recurrent intragenic *EGFR::EGFR*.E1E8 fusion that occurs in high-grade gliomas. The presence of EGFRvIII in other solid tumors has not been well characterized. We retrospectively reviewed advanced malignant solid tumor cases tested by a custom hybrid capture 610-gene next-generation sequencing platform from 2021 to 2022. EGFRvIII was identified in 17 of 4331 (0.4%) cases, including 16 of 238 (7%) brain tumors and 1/301 (0.3%) breast tumors. EGFRvIII-positive brain tumors were all glioblastoma IDH-wildtype, most with concurrent *TERT* promoter mutation (14 of 16), *EGFR* amplification (13 of 16), and *EGFR* mutation (8 of 16). The only EGFRvIII-positive breast lesion was a sarcomatoid neoplasm in a young female patient. A separate breast case tested outside our institution with reported EGFRvIII was noted in a young female patient with a malignant phyllodes tumor with stromal overgrowth. Microscopically, both EGFRvIII-positive breast tumors showed high-grade sarcomatoid morphology with brisk mitotic activity. In summary, EGFRvIII is rare, occurring primarily in glioblastoma and rarely in breast sarcomatoid neoplasm, with no instances identified in other tumor types in our series. This select group of patients may benefit from chemotherapy and/or targeted anti-EGFR therapy.

## 1. Introduction

Epidermal growth factor receptor (EGFR) is a transmembrane receptor tyrosine kinase from the ERBB protein family, which includes three other closely related receptors: ERBB2, ERBB3, and ERBB4 [1,2]. EGFR ligands include epidermal growth factor, transforming growth factor-α, and other growth factors and ligands [3]. Upon binding to extracellular ligands, EGFR forms homodimers or heterodimers with other ERBB family members and allosterically activates its intracellular receptor kinase domain [4]. The activation of EGFR initiates downstream intracellular signaling, leading to a range of effects, including controlled cell proliferation, differentiation, migration, adhesion, and apoptosis [5,6,7,8,9].

*EGFR* amplification and EGFR over-expression are observed in many cancer types, including glioblastoma [10,11,12], non-small-cell lung cancer [13], and breast cancer [14]. *EGFR* amplification occurs in approximately 50% of patients with primary glioblastoma as compared to 8% of patients with secondary glioblastoma and 5% of patients with non-small-cell lung cancer. *EGFR* amplification strongly correlates with increased EGFR expression and EGFR immunohistochemistry staining on tumor cells in glioblastoma [15,16]. Mutations in the *EGFR* gene are observed in many types of cancers; however, the recurrent mutated domains of *EGFR* in different cancer types vary. In non-small-cell lung cancer, the most common activating mutations are L858R in exon 21 and in-frame deletion in exon 19, and both of these exons are located within the intracellular kinase domain. In contrast, point mutations in the extracellular region of EGFR, such as R108K, A289V/D/T, G598D, and other extracellular domain mutations, are prevalent in glioblastoma. These extracellular domain mutants keep EGFR in an active conformation and are reported in 24% of glioblastomas [16].

The structural variants of EGFR, a group of intragenic EGFR fusions which resulted from the deletion of EGFR exons, are also frequently observed in glioblastoma. EGFR splicing variants in glioblastoma include EGFRvI (N-terminal deletion), EGFRvII (deletion of exons 14–15), EGFRvIII (in-frame deletion of exons 2–7), EGFRvIV (deletion of exons 25–27), and EGFRvV (deletion of exons 25–58); of these, EGFRvII and EGFRvIII have been confirmed to be constitutively active and oncogenic [17]. EGFRvIII occurs most commonly and is often associated as a late event in glioblastoma, occurring after amplification of wild-type *EGFR*. EGFRvIII lacks exons 2–7; the deletion of 267 amino acids within these exons creates a junction site with a new glycine residue, resulting in in-frame fusion of exons 1 and 8 and a tumor-specific epitope. EGFRvIII is constitutively activated and confers dysregulated intracellular EGFR signaling, leading to uncontrolled tumor cell growth and proliferation resistance to wild-type EGFR therapeutics [18]. EGFRvIII is associated with *EGFR* amplification in 30–40% of glioblastoma cases [19], and the expression of EGFRvIII is a negative prognostic marker for the overall survival in patients with glioblastoma surviving at least 1 year long [20,21]. In this context, specific treatments that directly target the EGFR pathway or activate the immune response against EGFRvIII have been recently developed and have been used in clinical trials as a single therapy or in combination with standard temozolomide treatment [22].

Although EGFRvIII has been well recognized in glioblastoma, the detection of EGFRvIII in the other solid tumors is not well established. Establishing a clear understanding of EGFRvIII distribution in various cancer types might be relevant for the selection of more effective target therapy. Next-generation sequencing (NGS) technology makes it feasible to rapidly and accurately identify EGFRvIII. Herein, we describe the distribution of EGFRvIII in 4331 samples from patients with diverse types of cancers.

## 2. Materials and Methods

### 2.1. Solid Tumor Cases

We retrospectively reviewed data from 4331 advanced solid malignant neoplasm cases at The University of Texas MD Anderson Cancer Center from 1 June 2021 to 31 August 2022 that were sequenced using the MD Anderson Mutation Analysis Precision Panel (MDA MAPP) to identify patients with detectable *EGFR::EGFR*.E1E8 intragenic fusion (EGFRvIII). Metaplastic carcinomas of the breast sequenced via MAPP were also included. All the patients provided written informed consent for their therapeutic procedure before beginning treatment and relevant data analysis in accordance with the institutional review board guidelines of MD Anderson, and the institutional review board approved this study. We collected data on the clinicopathological features of each case, including patient gender, age, tumor type, histology type, and mutation profile.

### 2.2. Immunohistochemistry

Immunohistochemical stains performed at MD Anderson used 4 μm thick formalin-fixed paraffin-embedded tumor tissue sections and Leica system (Leica Biosystems, -Deer Park, IL, USA) for CAM5.2 keratin (BD Biosciences, Becton, NJ, USA), GATA3 (Cell Marque, Rocklin, CA, USA), pancytokeratin cocktail (keratin AE1/3, keratin 8/18, CAM5.2, and MNF116), p63 (Biocare Medical, Pacheco, CA, USA), and TRPS1 (EPR16171, Abcam, Cambridge, UK).

### 2.3. Tissue Selection and DNA Extraction

The clinical molecular requests were reviewed by pathologists who selected the optimal tissue sample from the available formalin-fixed paraffin-embedded tissue and cytology smear. The pathologists marked the suitable areas to maximize the viable tumor-to-stromal ratio. The consecutive unstained tissue sections of 4 μm thickness were micro-dissected using the marked hematoxylin-and-eosin-stained slide as a guide. The minimum tumor percentage required for the sequencing analysis was 20%. The genomic tumor DNA was extracted and purified using AllPrep kit on a QIAcube liquid handling platform (Qiagen, Germantown, MD, USA) [23]. The peripheral blood DNA (a default source of germline DNA received in an EDTA tube) was extracted and purified using a Maxwell RSC Blood DNA Kit (Promega, Madison, WI, USA) [24].

### 2.4. Molecular Profiling Using Next-Generation Sequencing

The paired tumor DNA derived from the formalin-fixed paraffin-embedded tumor tissue or cytology sample and the control germline DNA derived from the peripheral blood sample were used for a mutational analysis in each patient. The minimum genomic DNA input was 50 ng. A mutational profile was performed using a clinically validated and laboratory-developed MAPP panel, a targeted hybridization capture-based NGS assay interrogating somatic variants (single nucleotide variants and insertion/deletion alterations) in 610 genes, amplifications in 583 genes, gene fusions in 34 genes, microsatellite instability, and tumor mutational burden. The targeted 2.1 megabases of the human genome was enriched with custom hybrid capture, 120nt double-strand DNA probes and sequenced on the NovaSeq 6000 NGS platform (Illumina, San Diego, CA, USA) using bidirectional paired-end sequencing. The whole-genome library construction included adapters carrying unique molecular indices. The original double-stranded DNA tagged with unique molecular indices allowed the statistical reconstruction of sequencing reads to be duplicated from a single-amplified genome.

Sequence alignment and analysis were performed inhouse via the MAPP bioinformatics pipeline, which relies on the dual-duplex molecular barcoding for consensus analysis to reduce sequencing artifacts and to increase sensitivity and positive predictive value. The post-variant calling analysis and annotation were performed using the laboratory-developed software OncoSeek (version 1.10.1.551). Sequence alignment was viewed with the Integrative Genomics Viewer (Broad Institute, Cambridge, MA, USA) using Human Genome Build 19 (Hg19) as the reference [25]. A minimum coverage depth of 100 unique molecular indices, error-corrected collapsed reads, was required (a minimum coverage of 100×); a somatic mutation is considered if a variant is absent in normal match control DNA and present in tumor sample with variant allelic frequency of 5% or higher. The synonymous somatic mutations in all coding regions, telomerase reverse transcriptase (*TERT)* promoter, and *TERC* non-coding RNA gene were reported. Inter-genic or intra-genic fusion was considered positive when 5 or more fusion breakpoint molecules were detected in at least 10 fusion molecules. Gene amplification was considered positive when the estimated copy number was 6 or higher. The tumor mutational burden was determined by measuring the number of somatic mutations occurring in sequenced genes and was specified as a rate (mutations per megabase (mut/Mb)). The microsatellite stability status was reported based on the analysis of at least 40 informative microsatellite loci.

## 3. Results

### 3.1. EGFRvIII in Solid Tumors

Of the 4331 cancers sequenced by the MAPP, we observed 17 (0.4%) cases that harbored EGFRvIII: 16 (7%) of 238 brain tumors; 1 (0.3%) of 301 breast tumors; and 0 (0%) of the 3792 malignancies of other types that originated from gastrointestinal, genitourinary, gynecologic, thoracic, skin, head and neck, and bone (Figure 1, Appendix A Appendix A). Among the 238 brain tumors, there were 112 glioblastomas, and the other 126 brain tumors predominantly comprised IDH-mutant astrocytoma and oligodendroglioma (Appendix A Appendix A). In 112 glioblastomas, 49 (44%) were *EGFR* amplified and 63 (56%) were *EGFR* not amplified. EGFRvIII was detected in 16/112 (14%) glioblastomas, 13/49 (27%) *EGFR* amplified glioblastomas, and 3/63 (5%) *EGFR* non-amplified glioblastomas. EGFRvIII was not detected in other brain tumors. The splice sites were detected within introns 1 and 7: one splice site per intron in nine cases and at least two spice sites per intron in the remaining eight cases. The co-existing *EGFR* mutations were present in eight cases, and the co-existing *EGFR* amplification was present in thirteen cases. The 17 EGFRvIII-positive cases all had a low tumor mutational burden and had a stable microsatellite status (Figure 2). The additional description of clinicopathological characteristics and detected genomic alterations for EGFRvIII-positive brain tumors and breast sarcomatoid tumors are listed in Appendix A Appendix A.

### 3.2. EGFRvIII-Positive Glioblastoma

The 16 patients with EGFRvIII-positive glioblastomas comprised six female and ten male patients; the patients’ ages at initial diagnosis ranged from 40 to 76 years, with a median of 59 years. The duration of clinical follow-up since diagnosis ranged from 3 to 22 months, with a median of 14 months. All 16 cases were IDH-wildtype primary glioblastoma, central nervous system World Health Organization grade 4. Microscopically, small-cell morphology was reported in four cases, of which three cases also showed concurrent *EGFR* amplification. Immunohistochemical staining for EGFR was performed in two cases, with both showing a diffuse strong expression of EGFR (Figure 3). The overall survival of patients with EGFRvIII-positive glioblastoma with reported small-cell morphology was worse than that of patients with EGFRvIII-positive glioblastoma without reported small-cell morphology (mean, 7.9 months vs. 17 months; Appendix A Appendix A).

Molecular profiling showed that all 16 cases were IDH-wildtype and histone H3 (H3)-wildtype and had copy number loss in chromosome 10. The copy number aberration in chromosome 7 was also noted in all 16 cases: 11 (69%) cases had *EGFR* amplification and chromosome 7 gain, two (13%) had *EGFR* amplification without chromosome 7 gain, and three (19%) had no *EGFR* amplification with copy number gain in chromosome 7. *EGFR* mutation was present in eight (50%) cases (Figure 2, Appendix A Appendix A). The most common co-existing mutation was a *TERT* promoter mutation in 14 of 16 cases: c.−124C > T mutation in eight cases, and c.−146C > T mutation in six cases. The second most common co-existing mutated genes included *PIK3CA, PTEN*, and *TP53,* the mutation of which were each present in four cases. Other commonly co-existing mutated genes were *PIK3R1, LRP1B,* and *PGR*, each in two cases.

### 3.3. EGFRvIII-Positive Breast Tumors

Only one of 301 breast lesions tested by the MAPP was positive for EGFRvIII (Figure 4), and this tumor did not exhibit *EGFR* amplification (Figure 2). EGFRvIII was not detected in the remaining 300 breast carcinomas. The EGFRvIII-positive breast lesion belonged to a young female patient (<30 years old) with a malignant high-grade sarcomatoid neoplasm with spindle and epithelioid cells, heterogenous architectural features, and focal chondromyxoid stroma. Some areas contained dilated ducts with clefting and large staghorn vessels. Numerous mitotic figures with geographic tumor necrosis were also identified. The immunohistochemical staining was not able to delineate the lineage (Appendix A Appendix A). A molecular study by the MAPP showed an EGFRvIII with a tumor mutational burden of 1 mut/Mb and stable microsatellite stability (Figure 4, Appendix A Appendix A). Somatic mutations in *TP53* and *PIK3CG* and amplification in *MYC, PTK2, RAD21, RECQL4,* and *RSPO2* were identified (Appendix A Appendix A). The patient received chemotherapy and mastectomy and was disease-free at follow-up 15 months after her initial diagnosis.

A separate case with EGFRvIII, which was not included in the MAPP cohort, was from a young female patient with the diagnosis of malignant phyllodes tumor with stromal overgrowth (Figure 5). The outside molecular testing report showed EGFRvIII, somatic mutations in *FYN, TP53,* and *TP53BP1*, and amplification in *EGFR*. The patient had surgery and was disease-free at her 12-month follow-up appointment.

We reviewed the remaining cases in this MDA MAPP cohort with the diagnosis or differential diagnosis of malignant sarcomatoid neoplasm of breast. We found four cases of metaplastic carcinoma, two with sarcomatoid/spindle, one with spindle/epithelioid, and one with adenosquamous differentiation. The patients’ ages ranged from 50 to 69 years, with a mean of 64 years. No patients had *EGFR* alteration (mutation, amplification, or fusion). The common somatic mutations were detected in *PIK3CA* (3), *TERT* (3), and *TP53* (2). *TERT* was mutated in two patients in the promoter (c.−124C > T) and in one patient in exon 2 (c.581G > A p.R194Q) (Figure 2, Appendix A Appendix A).

## 4. Discussion

Primary glioblastoma is a high-grade diffuse astrocytoma with elevated mitotic activity, microvascular proliferation, prominent intravascular fibrin microthrombi, and necrosis. It is typically IDH-wildtype and H3-wildtype. Frequent and diagnostically relevant molecular alterations in IDH-wildtype glioblastomas include *TERT* promoter mutations, *EGFR* gene amplification, and copy number gain in chromosome 7 combined with copy number loss in chromosome 10 (+7/−10 genotype) [26]. The presence of at least one of these aberrations in an IDH- and H3-wildtype diffuse glioma is sufficient for a diagnosis of molecular-characterized IDH-wildtype glioblastoma, even in the absence of morphologic evidence of microvascular proliferation and/or necrosis [27]. *TERT* encodes telomerase, which regulates telomere length during DNA replication and plays an important role in the senescence of normal somatic cells. Mutations in the *TERT* promoter region cause an upregulation of telomerase, which leads to telomere maintenance and oncogenesis seen in many tumor types, including glioblastoma [28]. Inhibitors that target telomerase activities are under investigation in phase 1/2 clinical trials.

*EGFR* is frequently altered in IDH-wildtype glioblastoma, with about 60% of tumors showing *EGFR* amplification, mutation, rearrangement, and/or altered splicing [16]. Among these alterations, the most frequent alteration is *EGFR* amplification, which has been observed in about 40% of all IDH-wildtype glioblastoma [26,29]. In most cases, *EGFR* amplification is associated with a second *EGFR* alteration, such as *EGFR::SEPT14* intergenic fusion, which is found in 4–8% of glioblastomas [30]; intragenic fusion EGFRvIII, which is found in 20–30% of glioblastomas [31,32,33]; and point mutations in the extracellular region of EGFR, which is found in 24% of glioblastomas [16]. The EGFRvIII testing in earlier studies was mostly performed via single gene/locus RNA-based reverse transcriptase polymerase chain reaction (PCR) and/or immunohistochemistry. In our study using multiplex NGS, we observed *EGFR* amplification in 44% of glioblastoma and EGFRvIII in 14% glioblastoma, similarly to previous studies.

Either the activating mutations of PIK3CA or inactivating mutations of PTEN in the phosphoinositide 3-kinase (PI3K) pathway could present as co-mutations with EGFRvIII of MAPK pathway in glioblastoma in this cohort study. The small-cell morphology is a subtype of glioblastoma presenting with a predominance of cells with highly monomorphic, small, round to slightly elongated, hyperchromatic nuclei and minimal discernible cytoplasm, with little nuclear atypia and brisk mitotic activity. It is reported that small-cell glioblastomas are uniformly IDH-wildtype and show *EGFR* amplification in ~70% of cases [27].

The correlation of small-cell morphology and EGFRvIII has not been reported. Lastly, whether EGFRvIII expression in tumors correlate with poor prognosis remains controversial. Shinojima et al. analyzed the prognostic value of *EGFR* gene amplification and mutation in 87 patients with newly diagnosed glioblastoma and demonstrated that EGFRvIII expression in the presence of *EGFR* gene amplification is an independent indicator and is the strongest indicator of a poor survival prognosis [34]. However, Felsberg et al.’s study, which included 106 glioblastomas with *EGFR* amplification, showed that EGFRvIII positivity was not associated with different progression-free or overall survival rates [33]. In this study, we showed the EGFRvIII-positive brain tumor cases were all primary glioblastoma, IDH-wildtype, and H3-wildtype with copy number loss in chromosome 10. Most cases also had *EGFR* amplification, *EGFR* somatic mutation, predominately in exons 6–9 located in the extracellular domain, and *TERT* promoter mutations. Multiple splicing sites of EGFRvIII were noted in half of the cases. Additionally, a subset of EGFRvIII-positive glioblastoma showed small-cell morphology, of which 75% of small-cell cases showed concurrent *EGFR* amplification. The small-cell morphology group had worse overall survival compared to patients without small-cell morphology.

Interestingly, we also identified one EGFRvIII-positive breast lesion without *EGFR* amplification or somatic mutation in the MAPP cohort and another separate case of malignant phyllodes tumor with reported EGFRvIII and *EGFR* amplification on testing performed outside our institution. Both cases occurred in young female patients, who were <30 years old at onset. Microscopically, both cases showed the morphology of high-grade sarcomatoid neoplasm and one case with typical phyllodes tumor area. In contrast, none of the four metaplastic carcinomas in this study had *EGFR* alteration.

The expression of EGFRvIII in breast neoplasms has been historically controversial. Earlier studies using nested PCR have reported the presence of EGFRvIII in breast carcinoma at variable frequencies via nested reverse transcription–PCR [35,36,37,38]. No EGFRvIII was detected in normal breast tissues [35]. However, the accuracy of reported EGFRvIII-positive rates in breast cancer and the possibility of clinically irrelevant/false-positive calls were difficult to assess with the relatively short/less specific primer designs and the inclusion of a second PCR in a nested PCR in these earlier studies. In contrast, EGFRvIII mRNA was not detected in Rae et al.’s study that used reverse transcription–PCR with a specific primer design for EGFR in any of the 55 breast cancer cell lines and 170 formalin-fixed paraffin-embedded primary breast cancer tissues [39]. Other types of EGFR intragenic fusion are also extremely rare, where only one breast cancer case (<0.1%) had reported EGFR intragenic fusion with an out-of-frame deletion of six exons [40].

In summary, results regarding the level of EGFRvIII expression and the proportion of breast carcinoma that express EGFRvIII might be subject to assay designs. Additional studies with larger sample sizes are needed to further investigate the role of EGFRvIII in malignant breast lesions, which may be a driver of the sarcomatoid morphology.

The differential diagnoses of the EGFRvIII-positive sarcomatoid breast case included in our analyses included metaplastic carcinoma, malignant phyllodes tumor with stromal overgrowth and primary sarcoma. Malignant phyllodes tumor and metaplastic carcinoma share morphologic similarities, especially when predominantly composed of spindle cells with no morphologically recognizable epithelial component, which poses a diagnostic challenge. Adequate sampling with multiple sections might aid in the identification of diagnostic features. However, sampling can be limited, especially on a small biopsy specimen. Immunohistochemical staining can help support the diagnosis of metaplastic carcinoma, such as diffuse expression of cytokeratins or p63/p40 in malignant spindle cells. However, the patchy staining of cytokeratin or p63/p40 has also been reported in the stromal component of malignant phyllodes tumors [41]. Recently, molecular studies provided some evidence of the genetic differences between malignant phyllodes tumors and metaplastic carcinoma of the breast. Pareja et al. studied the genetic profile of phyllodes tumors with or without a fibroadenoma-like area and found that *MED12* mutation at exon 2 was significantly more frequent in tumors with fibroadenoma-like areas; however, in tumors without fibroadenoma-like areas, the enrichment of cancer genes, especially *EGFR* mutation and amplification, was more frequently identified. No significant difference in the frequency of *TERT* genetic alteration was observed [42]. Gatalica et al. performed multiplex NGS on malignant phyllodes tumors and found that eight out of twenty-four (33%) cases had an overexpression of genes related to increased angiogenesis, especially *EGFR* amplification. The most common mutations included those involving *TP53* (50%) and *PIK3CA* (15%) [43]. Moreover, two *EGFR* alterations were identified, including EGFRvIII and *EGFR* V774M mutations. To date, our study includes the second case of malignant phyllodes tumor reported to harbor EGFRvIII, tested by NGS. Yet, metaplastic carcinoma consists of a group of heterogeneous tumors with distinctive morphology and intratumoral heterogeneity. Many studies have been performed to elucidate the molecular profiling of metaplastic carcinoma; however, specific pathognomonic mutations have yet to be identified. González-Martínez et al. studied the molecular profile of metaplastic carcinoma and found the commonly mutated genes to include *TP53* (58.7%), *PIK3CA* (32.8%), and *TERT* (29%), with genes involved in the PI3K pathway the most frequently identified in spindle-cell carcinoma. *MYC* (17.3%) was the most frequently amplified gene. The most common gene loss was *CDKN2A/CDKN2B* locus [44]. The tumor suppressor *TP53* encodes p53 protein, which regulates DNA repair, apoptosis, and cell cycle. Mutations in *TP53* have been reported to be significantly high in triple-negative, HER2-enriched, and metastatic breast carcinoma [45,46]. In our case of sarcomatoid breast neoplasm, NGS showed that besides EGFRvIII fusion, gene mutation involving *TP53* and *PIK3CG* and amplification of *MYC*, *PTK2*, *RAD21*, *RECQL4*, and *RSPO2* were also identified. In the case of malignant phyllodes tumor, somatic mutations of *TP53* and amplification of *EGFR* were identified in addition to EGFRvIII fusion. Additionally, the most common somatic mutations in the metaplastic carcinoma cases in our study included *PIK3CA*, *TERT*, and *TP53*, and none of the cases showed EGFR alterations. Our cases displayed a molecular profile that aligns with the findings reported by González-Martínez et al. [44].

We, respectively, reviewed a large cohort of advanced malignant solid tumor cases to identify EGFRvIII via NGS in the glioblastoma of the brain and sarcomatoid neoplasm of the breast. The cases were included in this cohort at the discretion of the treating oncologists based on patient status, tumor type, and tumor stage to search for potential therapeutic intervention. Thus, the sample size of each cancer type might be biased. Additionally, the detection of EGFRvIII by the MAPP can be potentially limited by its technical sensitivity and preanalytical factors, including and not limited to DNA degradation from formalin fixation and low tumor fraction. Limited by the small number of EGFRvIII-positive glioblastoma cases, we were unable to perform a meaningful survival analysis with statistical power in respect to the presence of small-cell morphology. Due to the rarity of sarcomatoid neoplasm of breast, we were unable to further delineate the diagnostic utility of EGFRvIII in differentiating malignant phyllodes tumors from metaplastic carcinoma and primary sarcoma of the breast, which overlap histologically and immunophenotypically. Future studies with specific focus on breast sarcomatoid neoplasms might help to elucidate the distinct molecular landscape of each entity.

## 5. Conclusions

In summary, intragenic EGFRvIII fusion in malignancies is rare, occurs primarily in glioblastoma, and only rarely occurs in breast tumors, with no instances identified in other tumor types in our cohort. Notably, the two breast lesions with EGFRvIII presented in young female patients and displayed sarcomatoid morphology similar to that of a previously described case of malignant phyllodes tumor with EGFRvIII. Additional studies with a larger sample size are needed to further investigate the role of EGFRvIII in malignant breast lesions, which may be a driver of sarcomatoid morphology. Given the emergence of agents such as pan-HER inhibitor, peptide, and chimeric antigen receptor T immunotherapy against EGFRvIII in clinical trials, this select group of patients may benefit from chemotherapy or targeted anti-EGFR therapy.

## Figures and Tables

**Figure 1 cancers-16-00006-f001:**
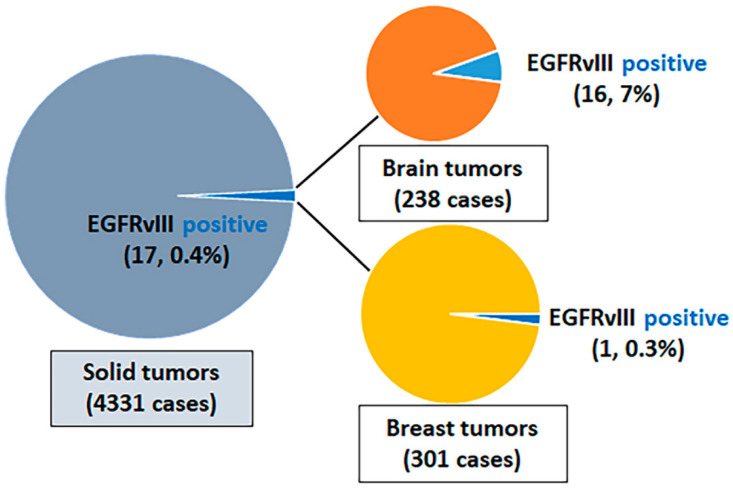
Frequency of EGFRvIII among solid tumors. EGFR, epidermal growth factor.

**Figure 2 cancers-16-00006-f002:**
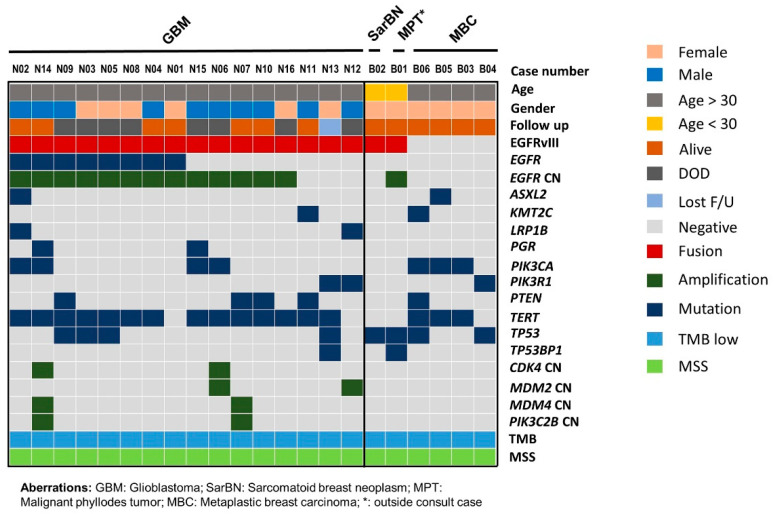
Mutation profiles of EGFRvIII-positive brain and breast neoplasms and metaplastic carcinoma of breast. Tumor type, patient age, gender, and last follow-up status are listed at the top. Detected recurrent mutated genes are shown below for each corresponding tumor type. GBM, glioblastoma; SarBN, sarcomatoid breast neoplasm; MPT, malignant phyllodes tumor; MBC, metaplastic breast carcinoma; CN, copy number; DOD, deceased of disease; F/U, follow up; TMB, tumor mutational burden; MSS, microsatellite stable.

**Figure 3 cancers-16-00006-f003:**
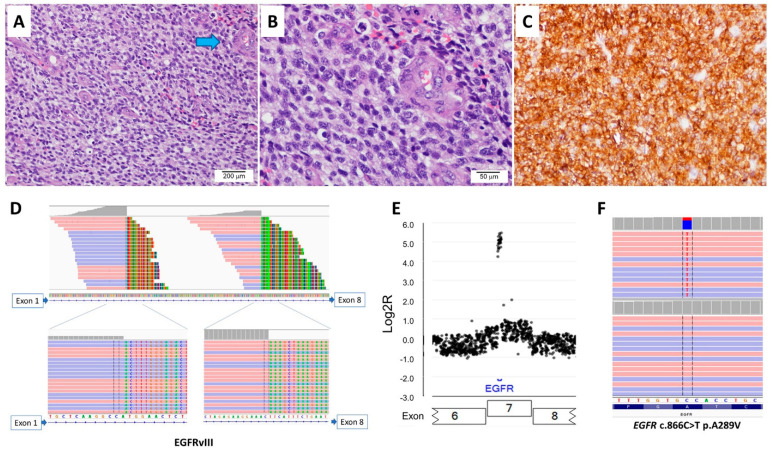
A case of IDH-wildtype glioblastoma with EGFRvIII and *EGFR* amplification. (**A**) A histological analysis shows a hypercellular lesion with microvascular proliferation (arrow in blue, 100×). (**B**) High power shows tumor cells with small-cell morphology (400×). (**C**) Immunohistochemical stain shows diffuse strong positivity for EGFR. The next-generation sequencing shows (**D**) two split sites within intron 1 of *EGFR*, (**E**) *EGFR* amplification, and (**F**) *EGFR* A289V mutation in extracellular domain.

**Figure 4 cancers-16-00006-f004:**
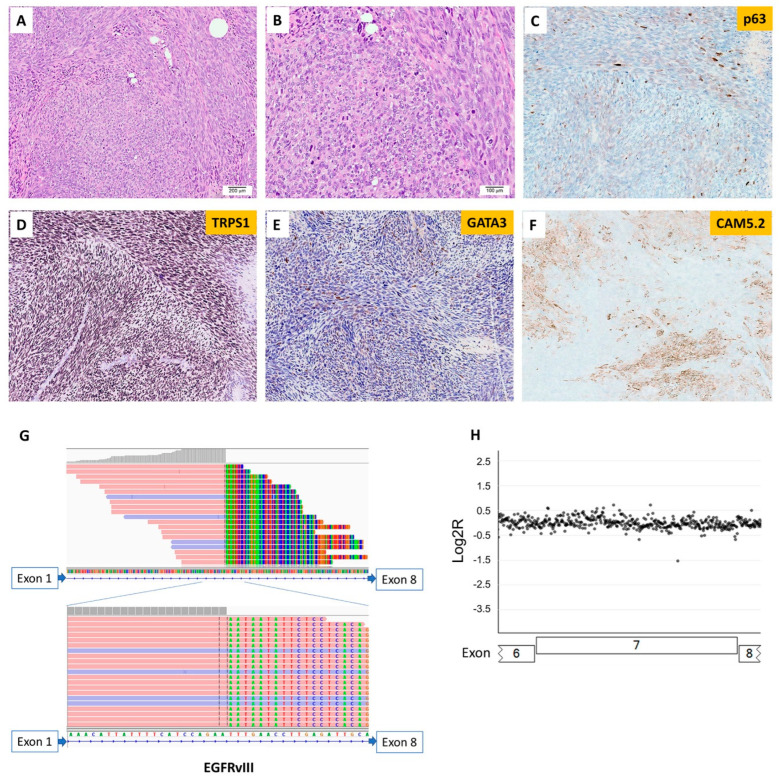
Sarcomatoid neoplasm of the breast with EGFRvIII. (**A**) Histology shows a malignant high-grade sarcomatoid neoplasm (100×). (**B**) Analysis under high-power magnification reveals tumor cells with spindle and epithelioid morphology and numerous atypical mitotic figures (200×). Tumor cells show focal expression of (**C**) p63, (**D**) diffusely positive staining for TRPS1, (**E**) positive staining for GATA3, albeit rarely, and (**F**) patchy expression of Cam5.2. The next-generation sequencing shows (**G**) one split site within intron 1 of *EGFR* and (**H**) no detectable *EGFR* amplification.

**Figure 5 cancers-16-00006-f005:**
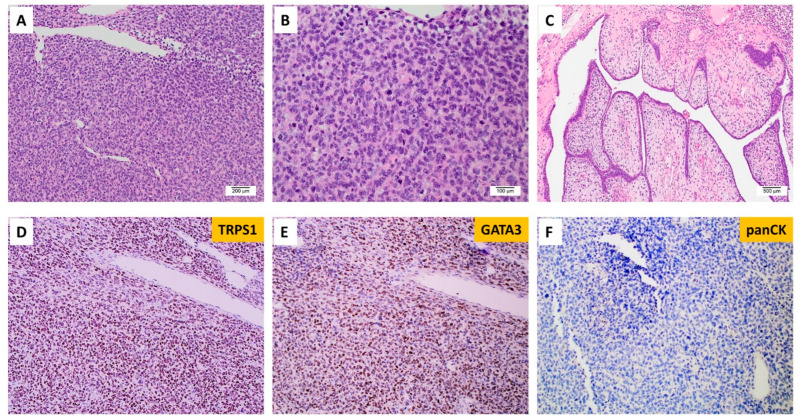
Malignant phyllodes tumors of the breast with EGFRvIII. (**A**) Histology shows a poorly differentiated high-grade neoplasm (100×). (**B**) Analysis under high-power magnification reveals pleomorphic epithelioid tumor cells with a high nuclear-to-cytoplasm ratio and no identifiable definitive epithelial component (200×). (**C**) A focal area with leaf-like fronds was identified (40×). Tumor cells show diffuse expression of (**D**) TRPS1 and (**E**) GATA3 and (**F**) rare expression of cytokeratin.

## Data Availability

The data presented in this study are available on request from the corresponding author. Only the de-identified data were used and the data are not publicly available due to protection of patient privacy.

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
