# Peer review of "Intragenic EGFR::EGFR.E1E8 Fusion (EGFRvIII) in 4331 Solid Tumors"

_cancers, 2023, doi:10.3390/cancers16010006_

Round 1

Reviewer 1 Report

Comments and Suggestions for Authors

The manuscript „Intragenic EGFR::EGFR.E1E8 Fusion (EGFRvIII) in 4331 solid tumors“ by Zheng et al studies the expression of an oncogenic form of the epidermal growth factor receptor termed EGFRvIII in 4331 solid tumors. The cohort studied here is fairly small. In some cases the tumor entities comprise just a few tumors. One would like to see the individual data in these cases. An expression of EGFRvIII in malignant glioma subtypes and rarely, in breast tumors was verified. Though interesting the study suffers from two major drawbacks. First of all, it is purely descriptive. Secondly, this type of study has been published repetitively before. At this stage it remains unclear what is new here and the authors should address the novelty and the correlation to previous studies. In general, the discussion is poor and little informative. In particular, studies on glioma should be discussed here as malignant phyllodes breast cancer as a rare and often benign tumor entity is less of an interest in the context of EGFRvIII. Further, one would like to see a critical comparison of the data obtained to previous studies.

Additional points:

- p4/line 171: 238 brain tumors: which ones?

-  p4/line 176: 112 glioblastomas: which percentage is EGFRvIII positive

- Figure 1: Panel should be expanded to illustrate subpopulations of glioma and breast tumors studied and their percentage of EGFRvIII-positivity.

- Malignant phyllodes breast cancer not in MAPP cohort. Why?

- p9/line 288; literature shows 20-30% EGFRvIII-positive glioma; EGFR-ampliefied?; comparison to current data?

- p12/line 397: Discuss status of EGFRvIII clinical trials

Author Response

Dear Editorial Board of Cancers:

We thank the reviewers and the journal for a thorough review of our manuscript “Intragenic EGFR::EGFR.E1E8 Fusion (EGFRvIII) in 4331 Solid Tumors” (MS ID: cancers-2545478). A revised version of our manuscript has been submitted. We removed a irrelevant reference previous #44 (page 11 line 416) and include 3 new references (new #28, 45, 46), therefore previous references #28-43 have been updated accordingly to new reference #29-44.

Below is our point-by-point response to the comments of the reviewers. 

  1. The cohort studied here is fairly small. In some cases, the tumor entities comprise just a few tumors. One would like to see the individual data in these cases.

RESPONSE:

  • We agree with the reviewer’s comment on the relatively small number of EGFRvIII positive cases, however this is large cohort study on a variety of solid tumors comprising of 4331 cases. Our study confirmed the rarity of EGFRvIII in non-glioblastoma solid malignancies. The detailed description of clinicopathological characteristics and genomic alterations for EGFRvIII positive brain tumors and breast sarcomatoid tumors are listed in supplementary tables S1 and S2, respectively. The somatic mutations of EGFRvIII positive brain tumors and breast sarcomatoid tumor cases are listed in new supplementary table S4 (previous supplementary table S3).
  • The additional description will provide important information about specific entities. We have included a notation for detailed description in Results page 4, lines 184-186.
  1. An expression of EGFRvIII in malignant glioma subtypes and rarely, in breast tumors was verified. Though interesting the study suffers from two major drawbacks. First of all, it is purely descriptive. Secondly, this type of study has been published repetitively before. At this stage it remains unclear what is new here and the authors should address the novelty and the correlation to previous studies.

RESPONSE:

  • We agree with the reviewer that this is a descriptive retrospective study, however this study has been performed on a wide variety of solid tumor cases including rare tumor types such as sarcomatoid neoplasm of breast. Additionally, this MAPP cohort was performed on the paired tumor and normal genomic DNAs by our custom designed next-generation sequencing (NGS) 610-gene MAPP panel using hybrid-capture, bidirectional paired-end sequencing, and adapters carrying unique molecular indices. The sequencing readout from MAPP panel includes EGFRvIII and other fusions, somatic mutations, amplifications, microsatellite instability and tumor mutation burden (See Material and Methods, Molecular profiling by next generation sequencing section in pages 3-4, lines 135-167 and Discussion in page 11 line 393).  The EGFRvIII testing in previous studies was mostly performed by single gene/locus RNA based reverse transcriptase (RT) PCR and/or relative non-specific immunohistochemistry in the earlier studies (references #33, 36, 39, 43, see Discussion, page 10, lines 299-301).  Consistent with previous findings, our study by NGS shows that EGFRvIII is rare in solid malignancies, occurring primarily in glioblastoma (see Discussion, page 10, lines 301-303). 
  • Prior to this study, only one case of EGFRvIII has been reported in breast neoplasm. The EGFRvIII detection method was fragment analysis sequencing using two sets of FAM linked primers to PCR amplify both the wild type and mutant EGFR alleles. The previously reported EGFRvIII positive case was a malignant phyllodes tumor, however patient age was not reported (reference #43). In a separate study using RT-PCR, EGFRvIII was not detected any of 170 breast cancer samples (0%, reference #39). EGFRvIII is intragenic in-frame deletion of exons 2-7 (see Introduction, page 2, line 80). An EGFR intragenic fusion with out of frame deletion of 6 exons was reported in a separate study (<0.1%, reference #40, see Discussion, page 10 line 345). By far, we reported the second and third cases of EGFRvIII positive breast neoplasm, one in sarcomatoid neoplasm and one in malignant phyllodes tumor, both cases in young female patients.

  1. In general, the discussion is poor and little informative. In particular, studies on glioma should be discussed here as malignant phyllodes breast cancer as a rare and often benign tumor entity is less of an interest in the context of EGFRvIII. Further, one would like to see a critical comparison of the data obtained to previous studies.

RESPONSE:

  • We appreciate the reviewer’s comment. We have included the comparison of EGFRvIII testing methods, reverse transcriptase PCR and/or immunohistochemistry in most of previous studies versus multiplex NGS in our MAPP cohort, to emphasize the novelty of this study. We also included the comparison of EGFRvIII positive rate in this study with previous studies (see Discussion, page 10, lines 294-297).
  • The malignant phyllodes tumor of breast can be not rare in large referral cancer center, and it is different from benign phyllodes tumor. Distant metastases are seen almost exclusively in malignant phyllodes tumor and metastasis can occur in nearly all internal organs. Patients with malignant phyllodes tumor have aggressive clinical course with mortality within 5-8 years, similarly as other malignant neoplasm including glioblastoma.

Additional points:

- p4/line 171: 238 brain tumors: which ones?

RESPONSE:

  • We appreciate the reviewer’s comment. The subpopulations of 238 brain tumors are included in the new Supplementary Table S3 and cited in Result (page 4, line 176). The previous Supplementary Table S3 has been changed to new Supplementary Table S4 (page 6, line 223; page 9, line 275).

-  p4/line 176: 112 glioblastomas: which percentage is EGFRvIII positive

RESPONSE:

  • Of 112 glioblastomas, EGFRvIII is detected by MAPP panel in 16 (14%) glioblastoma. The information has been included in Result section (page 4 lines 177-178) and the new supplementary Table S3.

- Figure 1: Panel should be expanded to illustrate subpopulations of glioma and breast tumors studied and their percentage of EGFRvIII-positivity.

RESPONSE:

  • We appreciate the review’s suggestion. We have created a new Supplementary Table 3 to include the suggested subpopulations of brain and breast tumors in this MAPP cohort and their EGFRvIII positive rate.

- Malignant phyllodes breast cancer not in MAPP cohort. Why?

RESPONSE:

  • The case of malignant phyllodes tumor of breast is an outside referral case with ancillary studies including next generation sequencing by a different panel performed at outside facility. Therefore, this case is not included in our MDA MAPP cohort.

- p9/line 288; literature shows 20-30% EGFRvIII-positive glioma; EGFR-amplified? comparison to current data?

RESPONSE:

  • We appreciate the reviewer’s suggestion. Previous studies show that ~40% of IDH-wildtype glioblastoma have amplified EGFR gene and about half of these tumors (or 20-30% of glioblastoma) express EGFRvIII (references #31-33). In our study, 44% of IDH-wildtype glioblastoma had EGFR amplification, 14% of IDH-wildtype glioblastoma had detected EGFRvIII by MAPP, comparable to previous literature reported findings. The suggested comparison has been included in discussion (page10, lines 301-303).

- p12/line 397: Discuss status of EGFRvIII clinical trials

RESPONSE:

  • The phase 2/3 clinical trials on therapies target to EGFRvIII in MAPK pathway are under investigation, predominantly in advance glioblastoma patients with EGFRvIII and/or EGFR These trials include EGFR inhibitor (Tarceva) in a phase 1/2 trial (NCT000301418), Chimeric Antigen Receptor T (CAR-T) immunotherapy targeting EGFRvIII in a phase 1/2 trial (NCT01454596, NCT03941626), a pan-HER irreversible inhibitor PF-299804 (Dacomitinib) in a phase 2 trial (NCT01520870), and an experimental cancer vaccine rindopepimut (also known as CDX-110) in phase 3 trial (NCT1498328).

         The suggested discussion has been included in Discussion (page12, lines 415-416).

Reviewer 2 Report

Comments and Suggestions for Authors

Zheng L et al. screened a total of 4331 solid tumors and identified the presence of EGFRvIII in 17 cases, 16 in brain tumors, and one in breast tumors. EGFRvIII occurs due to the deletion of six exons and the fusion of exon one to exon 8. EGFRvIII results in dysregulated intracellular EGFR signaling and uncontrolled growth of tumors. Authors conclude that EGFRvIII is a rare and primarily occurs in glioblastoma, and such patients could benefit from chemotherapy or targeted anti-EGFR therapy. This study highlights the rarity of EGFRvIII among solid tumors, but due to a small number of positive cases, it could not exhibit the significance of EGFRvIII in disease progression.

Specific points-

1.       Authors should highlight the lesions with arrows in Fig 3A.

2.       Authors have mentioned that the survival of patients with EGFRvIII-positive glioblastoma with small cell morphology was worse than the survival of EGFRvIII-positive glioblastoma without small cell morphology but did not any data to support this observation.

3.       For Fig 3C, it would be better to show an EGFR negative case sample along with the EGFR positive one.

4.       Authors show the presence of several mutations in 17 EGFRvIII positive cases such as TERT, PIK3CA, PTEN, and TP53 but did not discuss the potential physiological significance of these mutations in these cases.  

Author Response

Dear Editorial Board of Cancers:

We thank the reviewers and the journal for a thorough review of our manuscript “Intragenic EGFR::EGFR.E1E8 Fusion (EGFRvIII) in 4331 Solid Tumors” (MS ID: cancers-2545478). A revised version of our manuscript has been submitted. We removed a irrelevant reference previous #44 (page 11 line 416) and include 3 new references (new #28, 45, 46), therefore previous references #28-43 have been updated accordingly to new reference #29-44.

Below is our point-by-point response to the comments of the reviewer 2.

Zheng L et al. screened a total of 4331 solid tumors and identified the presence of EGFRvIII in 17 cases, 16 in brain tumors, and one in breast tumors. EGFRvIII occurs due to the deletion of six exons and the fusion of exon one to exon 8. EGFRvIII results in dysregulated intracellular EGFR signaling and uncontrolled growth of tumors. Authors conclude that EGFRvIII is a rare and primarily occurs in glioblastoma, and such patients could benefit from chemotherapy or targeted anti-EGFR therapy. This study highlights the rarity of EGFRvIII among solid tumors, but due to a small number of positive cases, it could not exhibit the significance of EGFRvIII in disease progression.

Specific points-

  1. Authors should highlight the lesions with arrows in Fig 3A.

RESPONSE:

  • We appreciate the reviewer’s suggestion. The arrows indicating microvascular proliferation have been included in Figure 3A (page 6). The legend for Figure 3A has been updated in page 6, line 212.
  1. Authors have mentioned that the survival of patients with EGFRvIII-positive glioblastoma with small cell morphology was worse than the survival of EGFRvIII-positive glioblastoma without small cell morphology but did not any data to support this observation.

RESPONSE:

  • Per reviewer’s suggestion, we include the survival graph in new Supplementary Figure S1 and update the citation in Result (page 6, line 208).
  1. For Fig 3C, it would be better to show an EGFR negative case sample along with the EGFR positive one.

RESPONSE:

  • We appreciate the comment from review. However, immunohistochemical staining for EGFR was only performed in 2 cases of glioblastoma with detected EGFRvIII in this cohort, and both cases showed diffuse strong expression of EGFR by immunohistochemistry study.
  1. Authors show the presence of several mutations in 17 EGFRvIII positive cases such as TERT, PIK3CA, PTEN, and TP53 but did not discuss the potential physiological significance of these mutations in these cases.  

RESPONSE:

  • We appreciate reviewer’s suggestion. The physiological significance has been included in discussion. The tumor suppressor TP53 encodes p53 protein, which regulates DNA repair, apoptosis and cell cycle. Mutations in TP53 are significantly high in triple-negative, HER2-enriched, and metastatic breast carcinoma (new references #45 and #46 are included, see Discussion, page 11, lines 382-385).
  • The full name of TERT, Telomerase Reverse Transcriptase, has been included in the first appearance in body of paper (page 4 line 160). TERT encodes telomerase, which regulates telomere length during DNA replication and plays an important role in the senescence of normal somatic cells. Mutations in TERT promoter region cause upregulation of telomerase, which leads to telomere maintenance and oncogenesis seen in many tumor types including glioblastoma (we include a new reference #28, Olymphios et al., Cancers (Basel), 2021, PMID: 33800183; previous references #28-43 have been updated accordingly to new reference #29-44). TERT promoter mutation is becoming one of the criteria in defining molecular characterized glioblastoma in gliomas lacking histologic features by 2021 WHO classification system. Inhibitors target to telomerase activities are under investigation in phase 1/2 clinical trials (See Discussion, page 9, line 286-291).
  • The activating mutations of PIK3CA or inactivating mutations of PTEN in the phosphoinositide 3-kinase (PI3K) pathway were co-mutations with EGFRvIII of MAPK pathway in glioblastoma in this cohort study. In advance glioblastoma patients, therapies target to EGFR and/or EGFRvIII in MAPK pathways, i.e. Chimeric Antigen Receptor T (EGFRvIII CAR-T), are under investigation in phase 1/2 clinical trials (see Discussion, page 10, lines 304-306).
  • In advanced triple negative breast cancer, PI3K inhibitors are under investigation in patient with PIK3CA activating mutation or inactivating PTEN mutation. In ER-positive HER2-negative breast cancer, PI3K inhibitors are preferred second-line therapy in patients with PIK3CA activating mutation per NCCN guidelines.

Sincerely,

Hui Chen MD, PhD and co-authors

The University of Texas MD Anderson Cancer Center

Pathology and Laboratory Medicine

1515 Holcombe Blvd

Houston, Texas 77030

Round 2

Reviewer 1 Report

Comments and Suggestions for Authors

None